# Environmental heterogeneity modulates the effect of plant diversity on the spatial variability of grassland biomass

Plant productivity varies due to environmental heterogeneity, and theory suggests that plant diversity can reduce this variation. While there is strong evidence of diversity effects on temporal variability of productivity, whether this mechanism extends to variability across space remains elusive. Here we determine the relationship between plant diversity and spatial variability of productivity in 83 grasslands, and quantify the effect of experimentally increased spatial heterogeneity in environmental conditions on this relationship. We found that communities with higher plant species richness (alpha and gamma diversity) have lower spatial variability of productivity as reduced abundance of some species can be compensated for by increased abundance of other species. In contrast, high species dissimilarity among local communities (beta diversity) is positively associated with spatial variability of productivity, suggesting that changes in species composition can scale up to affect productivity. Experimentally increased spatial environmental heterogeneity weakens the effect of plant alpha and gamma diversity, and reveals that beta diversity can simultaneously decrease and increase spatial variability of productivity. Our findings unveil the generality of the diversity-stability theory across space, and suggest that reduced local diversity and biotic homogenization can affect the spatial reliability of key ecosystem functions.

Understanding the mechanisms linking biodiversity with ecosystem stability is essential to anticipate the consequences of species loss for the sustainable delivery of critical ecosystem services[1–5]. Theory and empirical tests have demonstrated that plant biodiversity can stabilize the primary productivity of communities through time[4], and a variety of mechanisms have been proposed to explain this effect[6]. These mechanisms range from simple statistical relationships, such as the portfolio effect (i.e., statistical averaging of the independent and random fluctuations in the performance or abundance of different species[7]), to niche-based models like overyielding (i.e., increase of the mean productivity, relative to its variance, when a mixture exceeds the expected productivity based on monocultures[8]). Previous studies, nevertheless, identified asynchronous species responses to environmental fluctuations as the major underlying mechanism[9–12]. That is, biodiversity buffers productivity against environmental fluctuations, because reduced abundance of some species can be compensated for by increased abundance of other species[10,13]. Although this "insurance effect" is usually considered over time[13], theory suggests that it should also apply across space[13,14], because a larger species pool will be more likely to contain species that can grow well under different environmental conditions in space, decreasing the variability of productivity (i.e., increasing stability) across space[13,15]. Although the potential effect of biodiversity on the spatial variability of productivity has found some support in experimentally assembled communities[15–17] and natural systems[14], whether these results can be generalized is unknown and, to our knowledge, support for the different potentially involved mechanisms has not been evaluated empirically[13].

✉e-mail: pdaleo@mdp.edu.ar

Similar to its temporal counterpart, the spatial version of the insurance hypothesis[15,18] proposes stronger effects of plant biodiversity in heterogeneous environments compared to homogeneous environments[18,19] (see Fig. 1). This is because the greater the number of species present (i.e., either alpha or gamma-diversity), the higher the probability of including the set of best-performing species under different environmental conditions[18] (Fig. 1b). Despite these clear predictions for alpha- and gamma-diversity, the potential relationship between the spatial turnover in species composition (i.e., beta-diversity) and the spatial variability of productivity is harder to anticipate. Following the arguments above, as spatial turnover in species composition can emerge from (compensatory) changes among species under heterogeneous environmental conditions, increased beta-diversity may reduce spatial variability of productivity (Fig. 1c). However, changes in species composition can scale up to affect aggregate ecosystem properties, such as productivity[20] (especially if different species imply different functional traits). Under the spatial insurance theory[21], systems with high beta-diversity are expected to have high spatial variability in productivity across different patches at a given time, stabilizing productivity through time at larger spatial scales (that

integrate all patches)[21–24]. This positive effect of beta-diversity on the spatial variability of productivity may be especially important when patches are environmentally similar[21,25,26] (see Fig. 1c). Thus, the opposite predictions for the potential effect of beta-diversity on spatial variability of productivity can be reconciled if the outcome is context-dependent. Under low environmental heterogeneity, beta-diversity may mainly act as a destabilizing factor, because communities with different species compositions can respond differently to the common environment[21,27]. Under high environmental heterogeneity, in contrast, beta-diversity may act as a stabilizing factor, because different species may perform better under different environmental conditions (Fig. 1c). Biodiversity loss at different scales[5] is an important consequence of anthropogenic activities that also impacts the functioning of ecosystems. While biodiversity-functioning research has predominantly focused on temporal stability of biomass, less is known about spatial stability[13]. However, if biodiversity can buffer environmental change and stabilize spatial ecosystem functions and services, then biodiversity restoration and conservation will concurrently maximize functioning and spatial reliability[3] in changing conditions.

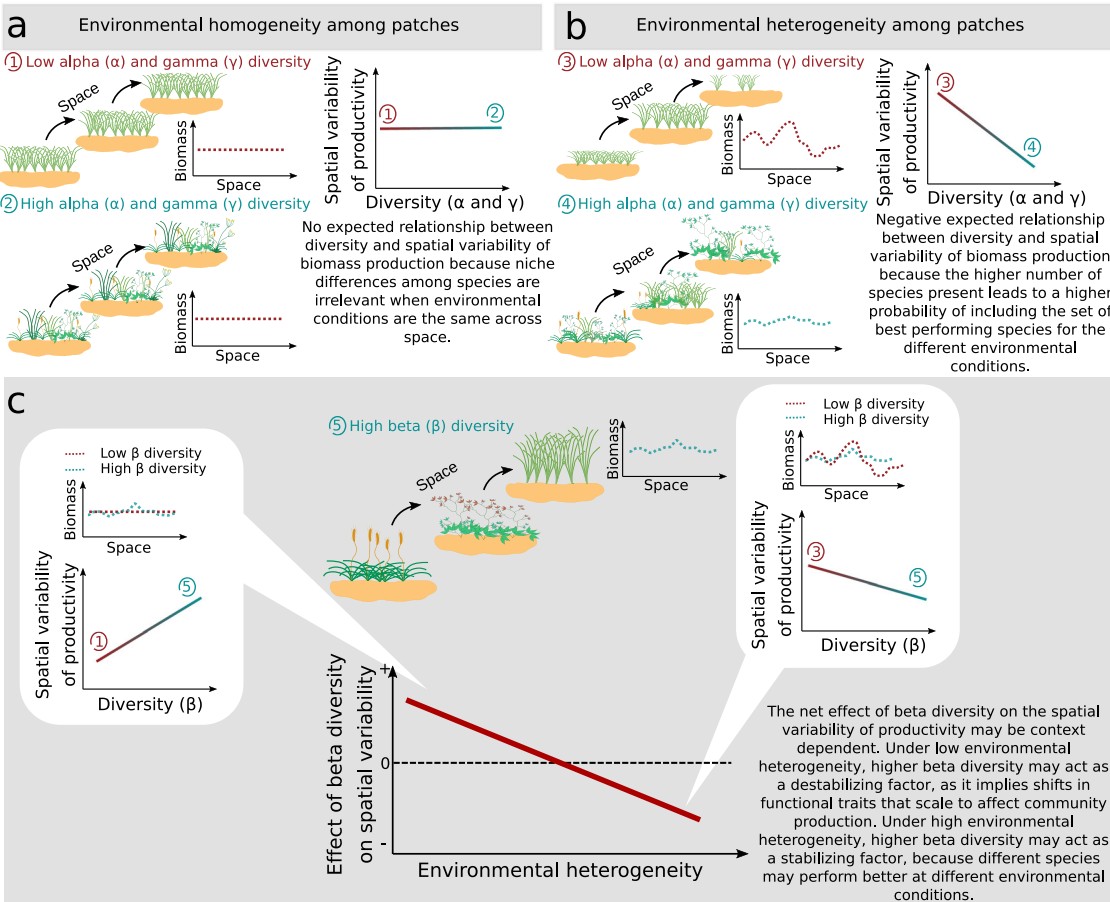

**Fig. 1 | Conceptual figure illustrating the effect of different scales of biodiversity on the spatial variability of aggregate ecosystem functions.** The insurance hypothesis postulates that biodiversity buffers aggregate ecosystem functions (e.g., biomass production) against environmental fluctuations, resulting in less variation within more diverse systems. This hypothesis was originally postulated for environmental fluctuations over time, but may also apply to spatial heterogeneity. **a** When environmental conditions are homogeneous, niche differences among species are non-important, and the variability of an aggregate ecosystem function is unaffected by alpha or gamma-diversity. **b** In contrast, in heterogeneous environments, different environmental conditions provide an array of niches. In this scenario, a species may be functionally insignificant under some environmental conditions, but more abundant or functionally important under other conditions. Thus, a highly diverse system may exhibit decreased variability of an aggregated ecosystem function compared to low diversity systems. In this scenario, a negative relationship is expected between alpha or gamma-diversity and the spatial variability of the function. **c** The net effect of beta-diversity on spatial variability of an aggregated ecosystem function may be context-dependent. When environmental heterogeneity is low, beta-diversity (that can be the result of priority effect or other stochastic processes) may act as a destabilizing factor as it can imply shifts in functional traits that scale up to affect community production. In contrast, when environmental heterogeneity is high, beta-diversity may act as a stabilizing factor because of niche complementarity.

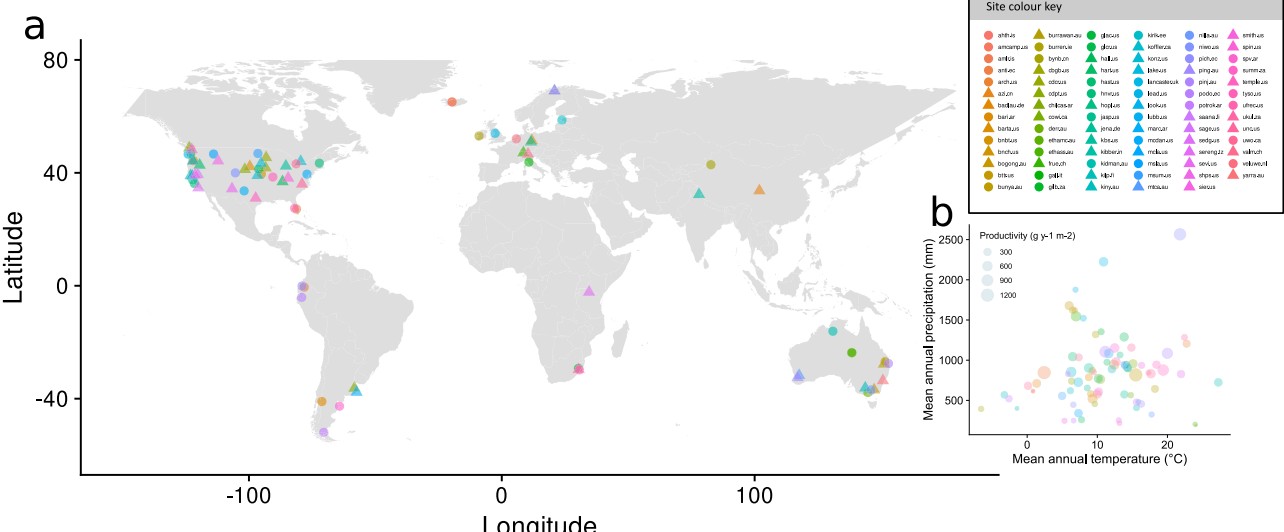

**Fig. 2 | Geographic and climatic distribution of grassland sites. a** Global map showing the locations of the 83 grassland sites included in this study. All sites were used to analyze diversity–variability relationships under ambient spatial environmental heterogeneity (pre-treatment conditions). Triangles denote the 42 sites that implemented the experimental protocol used to evaluate the effect of increased environmental heterogeneity on diversity–variability relationships. **b** The grassland sites span a wide range of mean annual productivity, mean annual temperature (MAT), and mean annual precipitation (MAP). Site color key shows the color assignment to each site, which is consistent in all figures.

Here, we explore the relationship between different scales of plant species diversity and spatial variability of productivity, measured as standing biomass, across 83 grasslands worldwide (see Fig. 2 and Supplementary Table 1) that are part of the Nutrient Network distributed experiment (NutNet; http://www.nutnet.org[28]). Using sets of ten unmanipulated plots ($25\,m^2$) arranged in blocks ($250\,m^2$) from these grasslands, we first analyzed whether local plot diversity (alpha-diversity), larger site-scale diversity (gamma-diversity), and among-plot variability in species composition (beta-diversity) are associated with the spatial variability of productivity, defined as the coefficient of variation (i.e., standard deviation/mean)[13,14] of aboveground standing biomass among plots. We also tested whether these associations are driven by two previously proposed niche-based mechanisms: (1) overyielding, or enhancing productivity (see ref.[29] for a temporal analog); and (2) insurance provided by spatial compensation between species[13]. Second, we tested how the association between different scales of diversity and spatial variability of productivity is affected by imposed spatial environmental heterogeneity. For this second objective, we used a subset of 42 grasslands that implemented a factorial nutrient addition and herbivore exclusion experiment[28] (see Fig. 2 and Supplementary Table 1). This experimental approach (see "Methods") represents a set of local plots ($25\,m^2$), with different resource supply, collectively representing a larger scale (an arrangement of 10 of those local plots resulting in $250\,m^2$) with spatial heterogeneity in environmental conditions (sampling methods and spatial scales are the same than for the previously described sampling; see Methods section). According to niche dimensionality theory[30,31], differences in resource supply and associated nutrient ratios should create patches with different niches and niche dimensions (i.e., different number of growth-limiting factors), increasing the spatial variability of productivity. Sites with high species diversity may have a greater probability of including the set of best-performing species in different patches (i.e., under different resource availability ratios), decreasing spatial variability of productivity[32]. Thus, environmental heterogeneity may increase variability of productivity across space and, in theory, alpha, beta and gamma-diversity may decrease this variability. We find that grasslands with higher alpha and gamma plant diversity have lower spatial variability of productivity as reduced abundance of some species are compensated for by increased abundance of other species. In contrast, grasslands with high beta-diversity have higher spatial variability of productivity. Furthermore, experimentally increased spatial environmental heterogeneity weakens the effect of plant alpha and gamma-diversity, and reveals that beta-diversity can simultaneously decrease and increase spatial variability of productivity.

## Results

### Global patterns of biodiversity-spatial variability of productivity relationships

Using unmanipulated (i.e., pre-treatment) data from the 83 grasslands, we found that alpha (linear mixed-effects models, $\chi^2 = 17.41$; $P < 0.001$) and gamma ($\chi^2 = 5.59$, $P < 0.05$) diversity were both negatively associated with spatial variability of productivity (Fig. 3a, b), whereas beta-diversity was positively associated with spatial variability of productivity ($\chi^2 = 9.77$, $P < 0.005$, Fig. 3c). We found no significant relationship between the different scales of biodiversity and the two separate components of spatial variability (i.e., $\mu$, the mean plot biomass; alpha: $\chi^2 = 0.52$; beta: $\chi^2 = 0.74$; gamma: $\chi^2 = 0.38$; all $P > 0.05$; Supplementary Fig. 1; and $\sigma$, the standard deviation of plot biomass; alpha: $\chi^2 = 0.03$; gamma: $\chi^2 = 1.29$; all $P > 0.05$; Supplementary Fig. 1), except for beta-diversity, that was positively associated with $\sigma$ (beta: $\chi^2 = 4.49$; $P < 0.05$; Supplementary Fig. 1). The patterns were consistent when modeled with type II regression (Supplementary Fig. 2) and for different diversity indices (Supplementary Table 2 and Supplementary Fig. 3). The patterns also persisted after accounting for differences in site environmental conditions, such as precipitation, temperature and seasonality (Supplementary Tables 3 and 4). Both alpha and gamma-diversity were negatively associated with species covariation, a spatial analog of species synchrony that (inversely) measures the degree of spatial biomass compensation between species (alpha: $\chi^2 = 33.43$, $P < 0.001$; gamma: $\chi^2 = 28.56$, $P < 0.001$; Fig. 3d, e). Species covariation was, in turn, strongly associated with spatial variability ($\chi^2 = 247.83$, $P < 0.0001$; Fig. 3g). However, we found no significant relationship between beta-diversity and species covariation ($\chi^2 = 2.31$, $P = 0.13$; Fig. 3f).

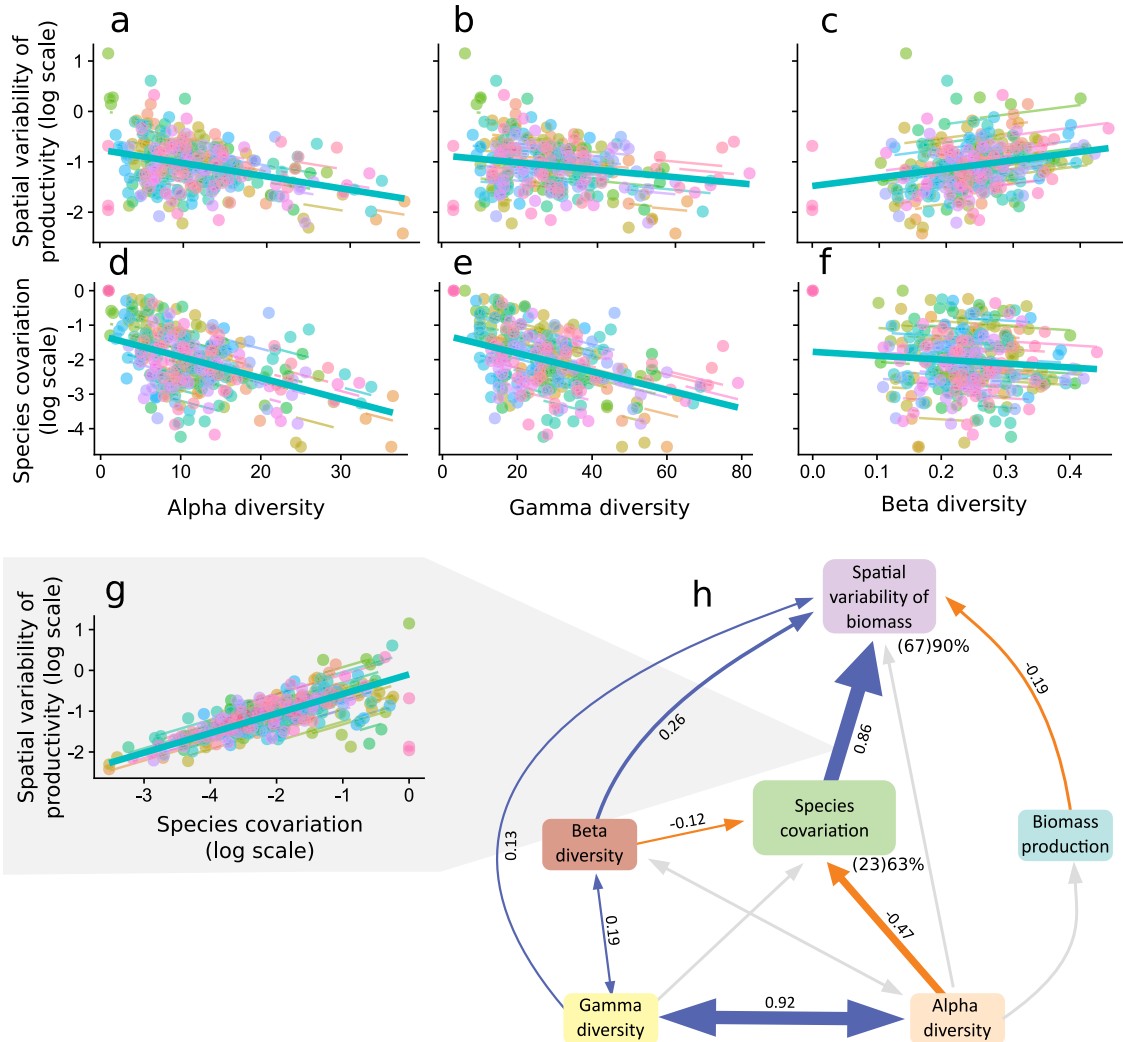

**Fig. 3 | The relationships between plant species diversity and spatial variability of productivity across 83 globally distributed grasslands sites of the Nutrient Network.** Both **a** alpha (slope and 95% CIs = −0.026 (−0.038 to −0.015)) and **b** gamma (−0.007 (−0.013 to −0.001)) diversity were negatively associated with the spatial variability. **c** Beta-diversity, in contrast, was positively associated with spatial variability (1.57 (0.59 to 2.54)); **d** alpha (−0.06 (−0.08 to −0.04)); and **e** gamma (−0.03 (−0.04 to −0.02)) diversity were negatively associated with species covariation. **f** Beta-diversity, in contrast, was not associated with species covariation (−1.27 (−2.92 to 0.38)). **g** Species covariation, in turn, was positively associated with spatial variability of productivity (0.48 (0.44 to 0.53)). **a**–**g** Different colors represent different sites (see Fig. 2 for site color key assignment), major lines (in turquoise) represent the fixed-effect linear regression slopes among sites and small colored lines show patterns within sites. **h** Structural equation model (SEM) analysis showing the direct and indirect pathways through which different scales of diversity determine the spatial variability of biomass. Model fit was assessed using

Shipley's test of d-separation (Fisher's *C* = 8.82, df = 6, *P* = 0.2). Solid blue arrows and solid orange arrows represent significant (*P* ≤ 0.05, no multiple comparison adjustments made) positive and negative paths, respectively, and light-gray arrows represent non-significant paths that were included in the initial model. Test of significance of path coefficients are two-sided for a difference from 0. Bidirectional arrows represent paths that were modeled as correlated errors (i.e., bidirectional relationships instead of causal and unidirectional relationships). Numbers next to the arrows are averaged effect sizes as standardized path coefficients; arrow widths reflect these standardized effect sizes. For spatial variability of biomass and species covariation, the marginal (i.e., explained by the fixed factors alone) and conditional (i.e., explained by both the fixed and the random factors; in parentheses) percent of variance explained is shown below and to the right of the variable name (see Supplementary Table 6 for non-standardized coefficient values and exact *P* values of individual paths).

## Direct and indirect effects of biodiversity on the spatial variability of productivity

To explicitly evaluate overyielding and compensatory changes between species[13] as mechanisms by which increased biodiversity could decrease spatial variability of biomass, we constructed a Structural Equation Model (SEM). The final model showed a good fit (Fisher's *C* = 8.82, df = 6, *P* = 0.2) and explained a high proportion of the total variance of spatial variability of productivity (marginal $R^2$ = 0.66; conditional $R^2$ = 0.90). Spatial variability of productivity was influenced primarily (and negatively) by species covariation (Fig. 3h). Higher alpha-diversity contributed to lower spatial variability through lower species covariation (Fig. 3h). Higher gamma-diversity also

contributed to lower spatial variability, but this effect was mainly because of a strong correlation with alpha-diversity (Fig. 3h). The indirect negative effect of gamma-diversity on spatial variability (through alpha-diversity) was partially offset by a direct positive effect (Fig. 3h). Higher beta-diversity, in contrast, contributed to spatial variability via two processes. First, beta-diversity positively contributed to spatial variability (Fig. 3h). Second, this positive effect was partially offset by a negative contribution of beta-diversity to spatial variability through lower species covariation (Fig. 3h). The model did not include pathways from any level of diversity to spatial variability mediated by biomass production (Fig. 3h), confirming the absence of overyielding in contributing to spatial variability seen in bivariate

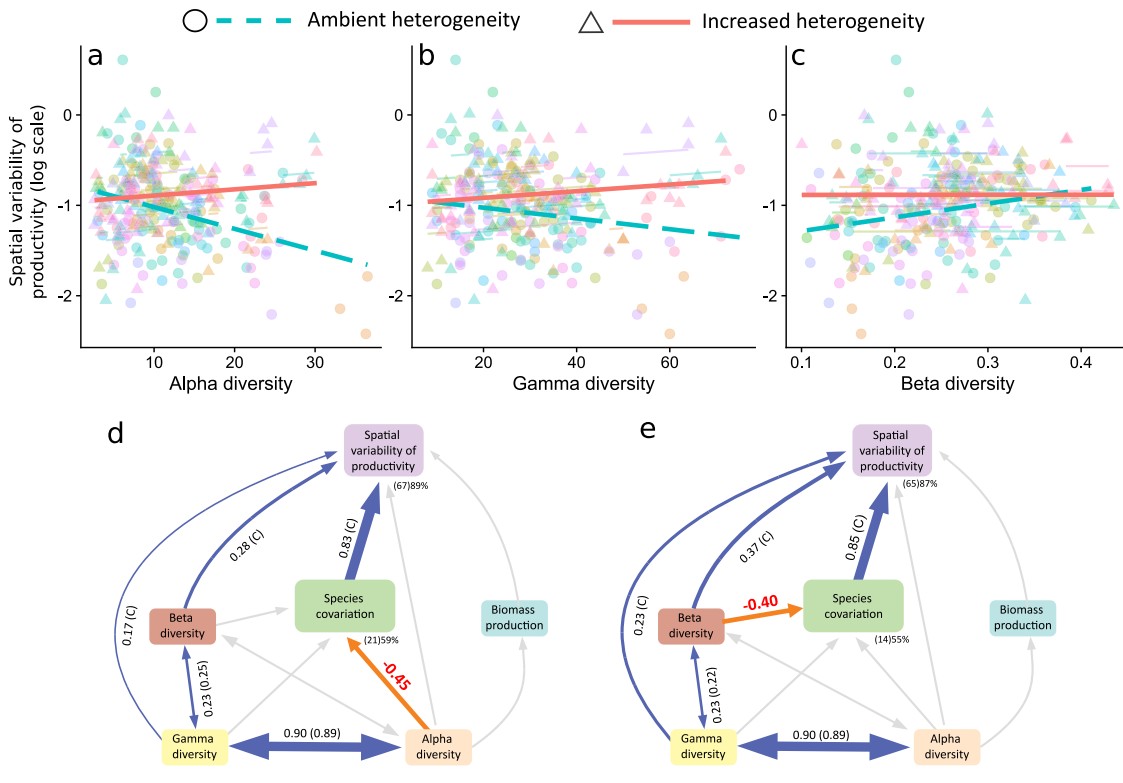

**Fig. 4 | Experimentally increased heterogeneity weakened the diversity-spatial variability relationships. a** Alpha-diversity (diversity*heterogeneity interaction slopes and 95% confidence intervals: 0.0046 (0.0077 to 0.0108)). **b** Gamma-diversity (0.0029 (0.0015 to 0.0043)). **c** Beta-diversity (−0.309 (−0.617 to −0.002)). Different colors represent different sites (see Fig. 2 for site color key assignment), major lines represent the fixed-effect linear regression slopes among sites and small colored lines show patterns within sites. Comparison of SEM models with **d** ambient and **e** experimentally increased spatial heterogeneity, using the subset of 42 sites that implemented the experimental protocol, identified two major changes (red numbers; $P \leq 0.05$ in multigroup analysis; see Supplementary Table 7 for exact $P$ values) in the pathways whereby increased heterogeneity weakened the three diversity–variability relationships: (1) the negative relationship between alpha-diversity and species covariation under ambient conditions was neutral under increased spatial heterogeneity; (2) the neutral relationship between beta-diversity and species covariation under ambient conditions became negative under increased spatial heterogeneity. Model fit was assessed using Shipley's test of d-separation (ambient heterogeneity: Fisher's $C = 1.108$, df = 6, $P = 0.981$; experimentally increased spatial heterogeneity: Fisher's $C = 3.108$, df = 4, $P = 0.54$). Solid blue

arrows and solid orange arrows represent significant ($P \leq 0.05$, no multiple comparison adjustments made) positive and negative paths, respectively (see Supplementary Table 8 for non-standardized coefficient values and exact $P$ values of individual paths), and light-gray arrows represent non-significant paths that were included in the initial model. Tests of significance of path coefficients are two-sided for a difference from 0. Bidirectional arrows represent paths that were modeled as correlated errors (i.e., bidirectional relations instead of causal and unidirectional relations). Numbers next to the arrows are averaged effect sizes as standardized path coefficients. Path coefficients that have been constrained (multigroup analysis; $P > 0.05$; see Supplementary Table 7 for exact $P$ values) are the same between the two models and are followed by a (C) (path coefficients are globally estimated, but standardized coefficients differ because the variance differs between groups, and thus the standardization). Numbers within brackets show bidirectional path coefficients estimated for the global model (i.e., as if they were conditional). Width of arrows reflects standardized effect sizes. The marginal (i.e., explained by the fixed factors alone) and conditional (i.e., explained by both the fixed and the random factors) percent variance of endogenous variables ($R^2$) are shown next to them (marginal between brackets).

relationships. After refitting the SEM using a smaller set of sites (54 sites in which soil samples were collected to include an estimation of spatial environmental heterogeneity), we found a positive direct effect of edaphic spatial heterogeneity on beta-diversity but the global model remained qualitatively unchanged (Supplementary Fig. 4).

## The effect of increased environmental heterogeneity

Next, we evaluated the effect of increased environmental heterogeneity on the relationship between spatial variability of productivity and species diversity using data from a subset of 42 grasslands (Fig. 2a) that experimentally enhanced environmental heterogeneity via nutrient and fencing treatments. Enhanced environmental heterogeneity increased the spatial standard deviation and the spatial variability of productivity, as well as beta-diversity (Supplementary Fig. 5). As experimental manipulation implied nutrient additions in most plots, it also increased μ, the mean plot biomass, and decreased alpha-diversity (Supplementary Fig. 5). However, enhanced environmental

heterogeneity did not affect species covariation or gamma-diversity (Supplementary Fig. 5). In addition, experimentally enhanced environmental heterogeneity flattened the relationships between the three scales of diversity and spatial variability (diversity*heterogeneity interaction, alpha: $\chi^2 = 23.41$; beta: $\chi^2 = 3.89$; gamma: $\chi^2 = 14.92$; all $P < 0.01$; Fig. 4a–c; see also Supplementary Fig. 6 for analysis including an intermediate level of heterogeneity).

Finally, using the data from the subset of grasslands that experimentally enhanced environmental heterogeneity, we refitted the SEM analysis, including experimentally increased spatial heterogeneity as a factor. Results identified two major changes in the pathways, compared to the model using data from unmanipulated (i.e., pre-treatment) plots, in which increased spatial environmental heterogeneity weakened the diversity–variability relationship for the three scales of diversity (i.e., there were two paths that varied between pre- and post-treatment; Fig. 4d, e). First, the negative relationship between alpha-diversity and species covariation under ambient conditions became

non-significant under increased heterogeneity (Fig. 4d, e). Second, the neutral relationship between beta-diversity and species covariation under ambient conditions became negative under increased heterogeneity.

## Discussion

Greater plant diversity is known to contribute to the decreased temporal variability of community productivity through higher asynchronous temporal dynamics among species in response to environmental fluctuations (species asynchrony[13]). Adding to this theory, we demonstrate that these same processes also occur through space. Across a wide range of global grasslands, spatial variability of site productivity declines with increasing plant diversity because of compensatory species responses to spatial heterogeneity (i.e., reduced species covariation across space). An obvious alternative explanation is that spatial environmental heterogeneity drives both spatial variability of productivity and biodiversity, but basic community theory predicts that more spatially variable environments should have higher biodiversity at both site (gamma) and local (alpha) scales due to niche-partitioning (increasing heterogeneity in environmental conditions promote species diversity by adding different niches)[33] and spatial mass effects (sink-source dynamics in which local species diversity can be enriched by species from the heterogeneous surrounding areas)[34]. Thus, this explanation would predict a positive association between biodiversity and spatial variability of productivity, contrary to the negative association we observed. In contrast to the observed decline in spatial variability of productivity with increasing alpha and gamma-diversity, greater beta-diversity was positively associated with spatial variability of productivity. These contrasting associations have been suggested by theoretical metacommunity studies (i.e., the spatial insurance theory)[21] that postulate that beta-diversity plays a key role in the temporal stability of productivity at regional scales, as it involves higher variation in temporal dynamics among local communities (spatial asynchrony), implying high spatial variability of productivity at a given time[13,21]. To our knowledge, nevertheless, this is the first study to provide empirical evidence. Finally, we demonstrate that spatial environmental heterogeneity, experimentally created by the addition of multiple types and combinations of nutrients and herbivore exclusions, increases (as expected) the spatial variability of productivity but weakens the relationships between different scales of plant diversity and this spatial variability.

The negative association of alpha and gamma-diversity with spatial variability of productivity can result from a combination of processes[6]. For instance, higher plant diversity often increases productivity (overyielding[35]). If this increase in the mean is not compensated by a proportional increase in its standard deviation, high-diversity sites should have lower spatial variability of productivity[8]. In contrast, as the effect of diversity on productivity may change along productivity gradients (shifting from positive in low-productivity communities to neutral or negative in high-productivity communities), diversity may decrease spatial variability by maintaining community productivity at intermediate levels (thus decreasing its standard deviation)[14]. In this study, both the bivariate relationships and the SEM analysis showed no significant direct relationship between diversity and the mean or the standard deviation of productivity when individually analyzed. But, when analyzing the spatial variability of productivity as a composite variable (i.e., coefficient of variation), our results suggest a combined effect on the two components (i.e., the ratio between standard deviation and mean productivity is a relative measure of variability that removes the impact of mean productivity). Results further suggest that the main underlying mechanism by which alpha and gamma-diversity decrease spatial variability of productivity is by decreasing species covariation (see also Supplementary Fig. S4). Different species can present non-correlated or negatively correlated changes in biomass production in different patches; thus, highly

diverse systems have lower spatial variability in aggregate productivity. Our results thus highlight the importance of compensatory species responses to environmental variation, as a general stabilizing mechanism for ecosystem function, not only in the temporal[4,23,36], but also in the spatial dimension as recently suggested[13].

The stabilizing mechanism of compensatory changes between species, contributing to more consistent biomass[6,13] may involve shifts in relative species abundances rather than abrupt compositional changes (i.e., species turnover), as our results show that large changes in species composition (i.e., high beta-diversity) are related to increases in the spatial variability of productivity. This pattern can arise because changes in species composition and spatial variability of productivity (or other aggregate functions) are both related to spatial heterogeneity in environmental conditions. The SEM analysis, nevertheless, only detected an indirect path between spatial environmental heterogeneity and spatial variability, a path that was mediated by beta-diversity. This suggests that at least part of the observed relationship between beta-diversity and spatial variability cannot be explained by its simultaneous correlation with environmental heterogeneity.

Experimentally imposed environmental heterogeneity weakened the bivariate negative relationship between spatial variability and both alpha and gamma-diversity on the one hand, and the bivariate positive relationship with beta-diversity on the other hand. Our SEM model suggests that this effect is due to a weaker relationship between alpha-diversity and species covariation. Thus, under experimentally increased environmental heterogeneity, biomass production of different species was no longer negatively correlated, i.e., they may have more coupled responses to spatial environmental variation, disabling the potential compensation between them. Our experimental design, in addition to the intended increased environmental heterogeneity (through varying combinations of nutrient additions), also led to higher mean plot biomass, and lower alpha-diversity as a consequence of increased mean nutrient inputs[37]. However, these effects should mostly affect variability rather than the relationships between diversity and variability as observed. Our SEM analysis also suggests that, under increased environmental heterogeneity, the weaker relationship between beta-diversity and spatial variability resulted from an enhanced negative contribution of beta-diversity to species covariation. Although of lower magnitude, this path was also detected using the full set of observational sites, but it was overcome by the stronger and positive direct path between beta-diversity and spatial variability. If different species are able to respond differently to environmental heterogeneity, higher dissimilarity in species composition among communities may decrease species covariation[13]. As this indirect path had a similar magnitude but opposite sign compared to the direct positive path, the two paths canceled each other out. Our results support theoretical work suggesting that beta-diversity acts as a destabilizing factor, as changes in species composition can involve shifts in functional traits that scale up to affect community production[20,21,26]. At the same time, beta-diversity can also act as a stabilizing factor, because different species may perform better under different environmental conditions[32]. When environmental variability is large enough, high contrast in environmental conditions drives coupled biomass covariation of shared species, but species divergence may partially offset this effect decreasing the spatial variability of productivity (Fig. 1c).

The most likely driver of spatial heterogeneity at the spatial scale of our study design (i.e., hundreds of meters) is plot-scale variability of biotic or abiotic conditions. Spatial heterogeneity in environmental conditions is usually the result of concurrent, superimposed gradients occurring at multiple spatial scales, or multiple disturbances interacting with each other[38]. Biomass production often varies in response to this combination of coarse and fine-scale heterogeneity. Results of studies evaluating the effect of biodiversity on ecosystem function are often scale-dependent. For example, small-scale studies are more

likely to be at the spatial scales at which niche-partitioning and competitive exclusion operate. Large-scale studies, on the other hand, are likely to detect the effects of site-scale factors (e.g., climate, herbivory) that may covary with diversity, thereby reducing the ability to detect niche-partitioning and competition[39]. At larger spatial scales, the importance of alpha-diversity may decrease (niche-partitioning becomes less important relative to extrinsic factors). Concurrently, the importance of beta-diversity may increase (as different species are filtered into environmental conditions where their traits most efficiently convert resources into biomass)[40]. Thus, even among the largest patches, diversity may continue to have an additional buffering effect on spatial variability in biomass production[41]. This natural spatial heterogeneity (even at small-scale) also contrasts with our experimentally increased heterogeneity, because our experimental landscape was characterized by high-contrast patches with sharp boundaries (i.e., clearly delimited experimental plots presenting within-plot homogeneous nutrient conditions and contrasting nutrient conditions among plots). Perhaps the clearest natural analogy takes place in some grazed systems, where a combination of abiotic (salinity, fire frequency, nutrients, water content) and biotic variables (grazer density, bioturbation, nutrient cycling) creates distinct patches of contrasting plant height, biomass and composition[42–44]. Those characteristics are also common features of some anthropogenic biomes (heterogeneous landscape mosaics, combining a variety of different land uses or land-use histories[45,46]) and similar to the management-driven landscape heterogeneity implemented to restore ecosystem complexity and diversity[47–50]. Thus, although the application of spatially variable management tools (such as patch-burning, patch-grazing, and land-use diversification) can increase spatial heterogeneity and restore diversity, they can potentially disrupt biodiversity-spatial variability relations.

Large-scale human impacts on ecosystems, such as land-use intensification, N deposition or species invasions, have been driving biotic homogenization, including losses in beta-diversity[51–54]. Our results suggest that those losses may lead to lower spatial variability in ecosystem-scale processes. The spatial homogenization in species composition may also imply higher spatial correlations in ecosystem temporal dynamics[21,23,25,55], increasing temporal variability of ecosystem functions at the landscape scale[21,23,56]. In addition, most of the drivers of biotic homogenization (e.g., eutrophication and trophic simplification[37]) also lead to reductions in alpha-diversity (but see ref. [57]). Thus, the potential loss of species at a local scale may still cause increased spatial (our results) and temporal[23,58] variability of ecosystem function, even in this biologically homogenized scenario. Biodiversity is thus a necessary prerequisite to ensure greater stability of key ecosystem functions in the face of an ever-expanding human footprint on environmental heterogeneity.

## Methods

To explore the relationship between different scales of plant biodiversity and spatial variability of productivity, we used observational (i.e., pre-treatment) data from 83 natural and semi-natural grassland ecosystems in 18 countries across 6 continents (see Fig. 2 and Supplementary Table 1) that are part of the Nutrient Network collaborative experiment (NutNet)[28]. All sites are dominated by herbaceous species, and together cover a wide range of grassland habitats that range from alpine grassland, to prairie, pasture, shrub-steppe, savanna, and old field. These grasslands also cover a wide range in elevation (0–4400 masl), mean annual precipitation (192 to 2566 mm yr$^{-1}$), mean annual temperature (−7 to 27 °C), latitude (52 degrees S to 69 degrees N), and aboveground productivity (0.5 to 1445 g m$^{-2}$ yr$^{-1}$; Fig. 2b). Study sites contained three replicate blocks each composed of ten 5 m × 5 m plots (see Supplementary Table 1 for exceptions). Here, we consider each plot as a "patch"[23], and the block of 10 plots as the "larger scale"[23]. Thus, each "larger scale" is composed of 10 "patches" (but see

Supplementary Table 1 for exceptions), and there are at least 3 "larger scales" per site, for a total of 83 sites, 271 "larger scales", and 2700 "patches". We defined alpha-diversity as species richness at the "patch" level, gamma-diversity as species richness at the "larger scale" level, and beta-diversity as the dissimilarity in species composition across the 10 "patches" within each "larger scale" (see details below).

To evaluate the effect of increased environmental heterogeneity on the relationship between spatial variability of productivity and species diversity, we used data from 42 of those sites (Fig. 2a) that implemented, for at least 4 years, an experiment with three nutrient addition treatments (nitrogen (N), phosphorus (P), potassium plus micronutrients (Kμ)), and vertebrate herbivore exclusion. At most sites plots were arranged in three blocks, each block containing the ten focal treatments: control (unfenced and unfertilized), +N, + P, +Kμ, +NP, +NKμ, +PKμ, +NPKμ, fenced (unfertilized), and fenced +NPKμ. Thus, each "larger scale" was composed of ten "patches" with different environmental conditions, that include variations in the availability of the most important limiting nutrients and variations in herbivory pressure. Here we used data from the 4th year of treatments. Nitrogen, P, and K were applied annually to experimental plots while micronutrients were applied just once, at the start of the experiment, to avoid toxic levels from overapplication. Nutrient addition rates and sources were: 10 g N m$^{-2}$ yr$^{-1}$ as timed-release urea ((NH$_2$)$_2$CO), 10 g P m$^{-2}$ yr$^{-1}$ as triple-super phosphate (Ca(H$_2$PO$_4$)$_2$), 10 g K m$^{-2}$ yr$^{-1}$ as potassium sulfate (K$_2$SO$_4$) and 100 g m$^{-2}$ yr$^{-1}$ of a micronutrient mix of Fe (15%), S (14%), Mg (1.5%), Mn (2.5%), Cu (1%), Zn (1%), B (0.2%), and Mo (0.05%). Fences were 2.1 m tall and excluded aboveground, non-climbing, vertebrate herbivores. The lower 0.9 m was composed of 10 mm woven wire mesh with a 0.3 m outward-facing flange stapled to the ground to exclude digging animals. The top 1.2 m was composed of five rows of wire. Minor variations in fence design are described in ref. [28]. Each plot was separated by at least 1.5 m from neighboring plots (1 m walkway and 0.5 m within-plot buffer), which served to minimize indirect effects of treatments in one plot on adjacent plots (for example, nutrient leaching, shading or mycelial networks). Although different sites started the experiment in different years, we used data from the 4th year of treatment implementation. Thus, sites have the same length of treatment years.

### Data acquisition and calculations

The variables described in this section were calculated separately for the pre-treatment and post-treatment (4th year of treatment implementation) sampling. Thus, we created two datasets, one based on pre-treatment (natural) conditions from 83 grasslands, and one with increased environmental heterogeneity from a subset of 42 grasslands.

We used aboveground live biomass as a surrogate measure of primary productivity. Aboveground live biomass was estimated destructively each year, at peak standing biomass, by clipping all aboveground biomass of individual plants rooted within two 0.1 m$^2$ (10 cm×100 cm) quadrats at ground level. Biomass was sorted into current (live and recently senescent material) and previous year's growth. For shrubs and subshrubs, all leaves and the current year's stems were collected. All biomass was dried to a constant mass at 60 °C prior to weighing to the nearest 0.01 g. Cover of each species was estimated non-destructively at a permanent 1 m × 1 m subplot within each plot. Plant species cover was recorded to the nearest 1% for each species in the plot. Cover was estimated independently for each species so that the total summed cover can exceed 100% for multilayer canopies. During pre-treatment sampling, soil samples were collected from each plot (three 25 mm diameter cores to 100 mm depth); because of missing samples, this dataset includes a subset of 54 of the 83 sites. Soils were air-dried to constant mass, weighed, and analyzed for pH, total carbon (C in %), total N (in %), P (ppm), and K (ppm) concentrations. C and N analyses were done at the University of Minnesota and the University of Nebraska via dry combustion GC analysis

(COSTECH ESC 4010 Element Analyzer) using cross-calibrated machines. Data also were generated on soil phosphorus, potassium, and micronutrients, soil pH, organic matter, and texture from each sample (A&L Analytical Laboratory, Memphis, TN, USA). Full details of Nutrient Network methods can be found in ref. [28].

We estimated alpha, beta, and gamma-diversity separately for each block, using the approach described in ref. [56]. Alpha-diversity was estimated as the mean plant species richness per plot across all plots within a block, whereas gamma-diversity was estimated as the total plant species richness within each block (i.e., block richness[56]). Beta-diversity was estimated as the Jaccard Dissimilarity Index across the ten plots within each block[56]; we calculated this index using the function "vegdist" from the "vegan" package[59] in R and then calculated the mean multivariate distance between the ten plots and their centroid using the function "betadisper" (also from the "vegan" package). We used these diversity indexes because they are all based on presence/absence. However, we evaluated if our results still hold using other common biodiversity indexes, such as the Shannon Entropy index, Inverse Simpson index and the Effective Number of Species (ENS) needed to reach the observed Probability of Interspecific Encounter ($S_{PIE}$)[60] for alpha and gamma-diversity as well as Whittaker´s multiplicative (i.e., alpha/gamma), additive (i.e., gamma−alpha) beta-diversity[61] and abundance-based multivariate beta-diversity (Supplementary Table 2 and Supplementary Fig. 3).

Stability is a multifaceted concept[62] that is commonly empirically measured as the inverse of variability (i.e., invariability);[63] the mean of an ecosystem property or function divided by its standard deviation. However, the term "stability" has a temporal connotation so, to avoid confusion, we defined spatial variability as the coefficient of variation (i.e., standard deviation divided by mean)[13,14]. Spatial variability of productivity was defined for each larger scale (i.e., block) as $\sigma/\mu$, where $\sigma$ is the spatial standard deviation of total live biomass, and $\mu$ is the spatial mean among the 10 plots of each larger scale. We estimated species covariation across space as a spatial analog of species synchrony (insurance effects may emerge from asynchronous species fluctuations[10,13,22]). It was measured for each block as:

$$Species\ covariation = \frac{\sigma^2}{\left(\sum_{i=1}^{S}\sigma_i\right)} \qquad (1)$$

where $\sigma^2$ is the variance in total plot live biomass, and $\sigma_i$ is the standard deviation of species i live biomass in a block with S species. Thus, if all species respond equally to spatial environmental variability, species covariation approaches 1, but if different species are capable of differently responding to this variability, species complement each other and species covariation approaches 0. As we do not have per-species biomass data, we used species' relative cover as a proxy. Cover of each species across the whole plot was multiplied by the total living biomass for the plot[36].

**Data analysis**
We first explored the relationship between different scales of biodiversity (i.e., alpha, beta and gamma-diversity) and the spatial variability of productivity using pre-treatment data from the 83 grasslands. We modeled these relationships with linear mixed-effects models using the "lmer" function in the "lme4" package[64] in R version 4.0.5 (R Core Team 2021). To improve normality, spatial variability was log-transformed before analysis. We used sites as random effects, allowing the intercepts and slopes of the regression to vary between sites if supported by model selection. We used a model-selection approach based on minimization of BIC following ref. [65], in which we compared models with and without a given random structure to determine which level of variation was required in the model. In all cases, model selection retained only variation among sites in the intercept. We also

modeled these relationships using type II regression (ranged major axis method) using the "lmodel2" package[66] in R to take into account the existence of sampling error of both predictor and response variables. As this model does not allow the inclusion of random structures (i.e., to reflect or multi-level design), we averaged values at the site level (i.e., instead of using three replicates per site, and to avoid pseudoreplication, we used the average value per site). To evaluate the two previously proposed niche-based mechanisms (i.e overyielding, which implies increases in the spatial mean of productivity as diversity increases, versus insurance, which implies decreases in species covariation as diversity increases), we also separately explored the relationship between biodiversity and each component of variability (i.e., $\sigma$ and $\mu$) and species covariation, using mixed-effects models as described above.

To remove the possible influence of key abiotic factors on the relationship between different scales of biodiversity and the spatial variability of productivity, we used a subset of bioclimatic variables representing (i) annual trends (mean annual temperature (°C) and precipitation (mm); seasonality (mean annual range in temperature (°C), the standard deviation in temperature, coefficient of variation of precipitation) and (ii) extreme or limiting environmental factors (mean temperature during the wettest 4 months (°C)). We performed a multiple regression of spatial variability against these climatic variables, kept the residuals, and then modeled the relationship between different scales of diversity and the obtained residuals, using type II regression. We also performed a multimodel inference (using the "MuMIn" package)[67] to select the simplest models that explained the most variation (of spatial variability) based on Akaike's information criterion (AIC). Candidate models represented every possible combination of explanatory variables (i.e., the subset of bioclimatic variables along with the different scales of diversity) and the interactions between bioclimatic variables and the different scales of diversity.

We then fit a Piecewise Structural Equation Model (Piecewise SEM)[68] to infer the direct and indirect effects of biodiversity on the spatial variability of productivity. Our model also aimed to explicitly evaluate whether increased biodiversity can decrease spatial variability of biomass production by the two previously proposed mechanisms (i.e., overyielding and decreased species covariation; see Supplementary Table 5). We began with a full conceptual model (see Supplementary Fig. 7) and followed a model simplification process in which non-significant paths were iteratively removed until only significant paths remained[69] and/or model fit was higher (i.e., minimization of BIC) than with further path removals. We incorporated site as a random effect in individual models[68] and model fit was assessed using Shipley's test of d-separation, which yields a Fisher's C statistic that is $\chi^2$ distributed[68]. In order to include an estimation of spatial environmental heterogeneity, we repeated the SEM analysis using the subset of 54 sites in which soil chemistry was measured. Environmental heterogeneity was estimated as the average Euclidean distance using the "vegan" package[59] in R for standardized soil parameters (soil C, N, P and K contents, and pH) and ambient light[56] among the ten plots within each block.

Lastly, we explored the effect of increased environmental heterogeneity using data from the 42 sites with experimental nutrient addition (see Supplementary Table 1). We first evaluated whether increased environmental heterogeneity affects the observed bivariate relationships between different scales of biodiversity and spatial variability of productivity, and then fitted the same SEM described above. For comparisons, we re-fit pre-treatment models for the subset of 42 experimental sites, and then performed a multigroup analysis to evaluate differences in path coefficients between pre- and post-treatment models using the "multigroup" function from the "piecewiseSEM" package[68] in R. In short, this analysis implements a model-wide interaction in which every term in the model interacts

with the grouping variable (i.e., pre- versus post-treatment). If the interaction is significant, then the path is free to vary by group; if not, then the path is constrained and takes on the estimate from the global dataset.

## Reporting summary

Further information on research design is available in the Nature Portfolio Reporting Summary linked to this article.

## Data availability

All data used in these analyses are publicly available on the Environmental Data Initiative (EDI) (https://doi.org/10.6073/pasta/583874460a0af70f93d3eee2f22f9a13); see ref. [70].

## Code availability

The complete R code supporting the findings of this study is freely available online at GitHub and archived through Zenodo (https://doi.org/10.5281/zenodo.7698668); see ref. [71].

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

## Acknowledgements

This work was generated using data from the Nutrient Network (http://www.nutnet.org) experiment, funded at the site-scale by individual researchers. Coordination and data management have been supported by funding to ETB and EWS from the National Science Foundation Research Coordination Network (NSF-DEB-1042132) and Long Term Ecological Research (NSF-DEB-1234162 and NSF-DEB-1831944 to Cedar Creek LTER) programs, and the Institute on the Environment (DG-0001-13). We also thank the Minnesota Supercomputer Institute for hosting project data and the Institute on the Environment for hosting Network meetings. Soil analyses were supported, in part, by USDA-ARS grant 58-3098-7-007 to ETB. This is KBS contribution #2344. NE and SH acknowledge the support of iDiv funded by the German Research Foundation (DFG– FZT 118, 202548816) and NE acknowledges funding by the DFG (Ei 862/29-1 and Ei 862/31-1). M.P. and R.T. were supported by the Estonian Research Council (PRG609), and the European Regional Development Fund (Centre of Excellence EcolChange). Y.L. is thankful to MPG Ranch for funding. A.J. acknowledges German Ministry for Education and Research (BMBF) for funding this research (SUSALPS; FKZ 031B0516C). N.G.S. acknowledges funding from the National Science Foundation (DEB-2045968) and Texas Tech University. Funding for the project comes from the European Research Council (ERC) under the European Union's Horizon 2020 research and innovation programme (grant agreement No. [101002987] to J.C.). This project was supported by grants from the Universidad Nacional de Mar del Plata (EXA1043/21), CONICET (PIP 2841) and ANPCyT (PICT 2019-3466) to O.I. and P.D.

## Author contributions

P.D., J.A., E.J.C., O.I., and Y.H. developed and framed research questions. P.D. and J.A. analyzed the data with contributions and inputs from E.J.C., O.I., Y.H., E.T.B., J.D.B., E.W.S., A.S.M., S.W., P.M.T., and S.B. P.D. and J.A. wrote the paper with contributions and inputs from E.J.C., O.I., P.M.T., J.D.B., E.T.B., M.B., A.S.M., J.P., M.S., E.W.S., S.W., S.B., L.A.B., J.A.C., C.R.D., T.L.D., I.D., N.E., D.S.G., S.H., A.J., J.M.H.K., Y.L., R.L.M., J.L.M., B.M., T.O., M.P., P.L.P., S.A.P., A.C.R., C.R., N.G.S., C.S., R.T., G.F.V., P.A.W., and Y.H. E.W.S., E.T.B., and P.A.W. are Nutrient Network coordinators.

## Competing interests

The authors declare no competing interests.

## Additional information

Pedro Daleo [1] ✉, Juan Alberti[1], Enrique J. Chaneton[2,37], Oscar Iribarne[1], Pedro M. Tognetti [2], Jonathan D. Bakker [3], Elizabeth T. Borer [4], Martín Bruschetti[1], Andrew S. MacDougall[5], Jesús Pascual[1], Mahesh Sankaran [6,7], Eric W. Seabloom [4], Shaopeng Wang [8], Sumanta Bagchi[9], Lars A. Brudvig[10], Jane A. Catford [11,12], Chris R. Dickman [13], Timothy L. Dickson[14], Ian Donohue[15], Nico Eisenhauer [16,17], Daniel S. Gruner [18], Sylvia Haider [16,19,20], Anke Jentsch[21], Johannes M. H. Knops[22], Ylva Lekberg [23], Rebecca L. McCulley [24], Joslin L. Moore[12,25,26], Brent Mortensen[27], Timothy Ohlert[28], Meelis Pärtel [29], Pablo L. Peri[30], Sally A. Power [31], Anita C. Risch [32], Camila Rocca[1], Nicholas G. Smith [33], Carly Stevens [34], Riin Tamme [29], G. F. (Ciska) Veen [35], Peter A. Wilfahrt[4] & Yann Hautier [36]

[1]Instituto de Investigaciones Marinas y Costeras (IIMyC), UNMDP—CONICET, CC 1260 Correo Central, B7600WAG Mar del Plata, Argentina. [2]IFEVA-Facultad de Agronomía, Universidad de Buenos Aires—CONICET, Av San Martín 4453 C1417DSE, Ciudad Autónoma de Buenos Aires, Argentina. [3]School of Environmental and Forest Sciences, University of Washington, Seattle, WA 98195, USA. [4]Department of Ecology, Evolution & Behavior, University of Minnesota, St. Paul, MN 55108, USA. [5]Department of Integrative Biology, University of Guelph, Guelph, ON N1G2W1, Canada. [6]National Centre for Biological Sciences, Tata Institute of Fundamental Research, Bengaluru, Karnataka 560065, India. [7]School of Biology, University of Leeds, Leeds LS2 9JT, UK. [8]Institute of Ecology, College of Urban and Environmental Science, and Key Laboratory for Earth Surface Processes of the Ministry of Education, Peking University, 100871 Beijing, China. [9]Centre for Ecological Sciences, Indian Institute of Science, Bangalore, Karnataka 560012, India. [10]Department of Plant Biology and Program in Ecology, Evolution, and Behavior, Michigan State University, East Lansing, MI 48824, USA. [11]Department of Geography, King's College London, 30 Aldwych, London WC2B 4BG, UK. [12]School of Ecosystem and Forest Sciences, University of Melbourne, Parkville, VIC 3010, Australia. [13]Desert Ecology Research Group, School of Life & Environmental Sciences, University of Sydney, Camperdown, NSW 2006, Australia. [14]University of Nebraska at Omaha, Department of Biology, Omaha, NE, USA. [15]Zoology, School of Natural Sciences, Trinity College Dublin, Dublin 2, Ireland. [16]German Centre for Integrative Biodiversity Research (iDiv) Halle-Jena-Leipzig, Leipzig, Germany. [17]Institute of Biology, Leipzig University, Leipzig, Germany. [18]Department of Entomology, University of Maryland, College Park, MD 20742, USA. [19]Institute of Biology/Geobotany and Botanical Garden, Martin Luther University Halle-Wittenberg, Halle, Germany. [20]Institute of Ecology, Leuphana University of Lüneburg, Lüneburg, Germany. [21]Disturbance Ecology, BayCEER, University of Bayreuth, 95447 Bayreuth, Germany. [22]Department of Health & Environmental Sciences, Xi'an Jiaotong-Liverpool University, Suzhou, Jiangsu, China. [23]MPG Ranch and University of Montana, W.A. Franke College of Forestry and Conservation, Missoula, MT 59812, USA. [24]Department of Plant and Soil Sciences, University of Kentucky, Lexington, KY 40546, USA. [25]Arthur Rylah Institute for Environmental Research, 123 Brown Street, Heidelberg, VIC 3084, Australia. [26]School of Biological Sciences, Monash University, 25 Rainforest Walk, Clayton, VIC 3800, Australia. [27]Department of Biology, Benedictine College, Atchison, KS, USA. [28]Department of Biology, Colorado State University, Fort Collins, CO, USA. [29]Institute of Ecology and Earth Sciences, University of Tartu, Tartu, Estonia. [30]Instituto Nacional de Tecnología Agropecuaria (INTA)- Universidad Nacional de la Patagonia Austral (UNPA) -CONICET. Río Gallegos, Santa Cruz, Argentina. [31]Hawkesbury Institute for the Environment, Western Sydney University, Locked Bag 1797, Penrith, NSW 2751, Australia. [32]Swiss Federal Institute for Forest, Snow and Landscape Research WSL, Community Ecology, Zuercherstrasse 111, 8903 Birmensdorf, Switzerland. [33]Department of Biological Sciences, Texas Tech University, Lubbock, TX 79409, USA. [34]Lancaster Environment Centre, Lancaster University, Lancaster LA1 4YQ, UK. [35]Department of Terrestrial Ecology, Netherlands Institute of Ecology, PO Box 50, 6700 AB Wageningen, The Netherlands. [36]Ecology and Biodiversity Group, Department of Biology, Utrecht University, Padualaan 8, 3584 CH Utrecht, The Netherlands. [37]Deceased: Enrique J. Chaneton. ✉e-mail: pdaleo@mdp.edu.ar

