## [Peer Review File · Nature Communications]

REVIEWER COMMENTS

Reviewer #1 (Remarks to the Author):

This paper uses data from a large global study to examine relationships between plant diversity and productivity across local spatial scales. The authors test whether overyielding or spatial compensation are the mechanisms underlying the diversity – productivity relationships. Overall this is a very interesting paper using a strong dataset to test an important question. The analytical approaches used in the paper are reasonable and the paper is very well written. I have a few fairly minor comments below.

L80-81. This sentence is hard to follow as reduced diversity can be correlated with homogenization.

L153 and on. Some symbology incorrectly rendered in the pdf

Figure 1. The cursive axis labels are indistinct and difficult to read. Please use a standard font.

Figure 4. Could the switch from an alpha diversity mechanism to a beta diversity mechanism with nutrient addition be an artifact caused by the differences in scale? I would imagine that the 250m scale for the second experiment would have greater beta diversity which could influence the relative importance of the alpha vs. beta diversity in the SEM. In addition, figure S6 appears to show that adding nutrients increased beta diversity and lowered alpha diversity which could also explain the shift in relationships between the two SEM models.

Reviewer #2 (Remarks to the Author):

In this manuscript, the authors tested the relationships between different levels of biodiversity (alpha, beta, and gamma diversity) and spatial variability in productivity and examined how increased environmental heterogeneity can alter these relationships. To do this, the authors used the dataset of a global network of experimental grasslands in which nutrients and herbivore exclusion were manipulated. They found that alpha and gamma diversity can reduce overall spatial productivity variability. They also found that beta diversity could either increase (when environmental heterogeneity is low) or decrease (when environmental heterogeneity is high) spatial variability in productivity. The study was well designed, and the statistical analysis used by the authors (LMMs and SEMs) and the interpretation of the results seems good to me. I have only a few minor comments that I hope can clarify the manuscript.

- In the title, "cancel" is a very strong word; I suggest replacing it with a weaker one. What "diversity" (alpha, beta, or gamma) are you referring to in the title, since increased heterogeneity has different effects on different scales of diversity?
- I feel the introduction missed to tell readers why we should care about spatial variability. Does it affect ecosystem functioning or services when human activities increase or decrease spatial variability (productivity)?
- Both environmental heterogeneity and beta-diversity are spatially scale dependent. The experimental systems used in this study are still at the plot level (or an group of plots). The spatial change in both environmental heterogeneity and beta diversity across larger natural spatial scales might be very different from the system studied here. The author should also discuss the generality of the results at larger spatial scales. Discussing the scale of the space across which the study was performed is important as the authors also linked their results with biotic homogenization, which is also a spatial scale-dependent process.
- In the study, increased environmental heterogeneity is treated as a binary variable (i.e. ambient heterogeneity and increased heterogeneity). The grouping is very coarse. The authors should also try to quantify the effect of environmental heterogeneity, maybe by grouping the blocks to different levels of environmental heterogeneity. I can of course understand that the number of the levels will be limited by sample size.
- L334: why 2750 patches from 271 larger scales?
- L341-343: I suggest here more explicitly describe the 10 patches (treatments) with +N, +P, +Fence etc as described in Borer et al. 2014.
- The grasslands likely differ in the length of years across which they are sampled. Although this is more likely to have a consequence on temporal variability of the plots. But have the authors considered that this also has some consequences on the spatial variability. With increased sampling period, the plots in the same grassland are more likely to be affected by external climatic drivers (extreme climates for example) and homogenize the plots.
- L369-370 is repeating L365. Consider simplifying it.
- L390: I do not agree that the three diversity indexes are independent. Gamma diversity is somehow a product of alpha and beta diversity.
- L425: I am not familiar with this regression. But what do you mean by averaging values? Can you more explicitly specify this?
- L437-439: Instead of doing this way, why not just include both climatic variables and biodiversity as explanatory variables in the same model? And the authors can also check whether there is any interactive effect between climatic variables and diversity indexes.
- L448: "when appropriate" sounds very unclear. Please specify it.

- The authors should also deposit and share the codes and core dataset in one of public repositories, so the interested referee/readers can reproduce the results.
- Fig. S1: why the authors hypothesize that there is no effect of environmental heterogeneity on gamma diversity.
- L455: Do you mean Table S1 here?

Instituto de Investigaciones Marinas y Costeras
Consejo Nacional de Investigaciones Científicas y Técnicas

CONICET

Laboratorio de Ecología

UNMDP

CC 1260 Correo Central, 7600 Mar del Plata, Argentina (B7600WAG)

Tel. 54 0223 475 3554, FAX: 54 0223 4753150

Lab. web page: <http://www.ecologia.mdp.edu.ar/english/index.htm>

Point-by-point responses to reviewers comments (authors' replies in blue, line numbers correspond to the file with track changes)

Reviewer #1 (Remarks to the Author):

This paper uses data from a large global study to examine relationships between plant diversity and productivity across local spatial scales. The authors test whether overyielding or spatial compensation are the mechanisms underlying the diversity – productivity relationships. Overall this is a very interesting paper using a strong dataset to test an important question. The analytical approaches used in the paper are reasonable and the paper is very well written.

[Re]: We appreciate these positive comments about our paper.

I have a few fairly minor comments below.

L80-81. This sentence is hard to follow as reduced diversity can be correlated with homogenization.

[Re]: Thank you for pointing this out. We edited the sentence to give it more clarity (see lines 86-88).

L153 and on. Some symbology incorrectly rendered in the pdf

[Re]: We indeed noticed that the symbol chi squared was incorrectly rendered in the pdf. We have now fixed it.

Figure 1. The cursive axis labels are indistinct and difficult to read. Please use a standard font.

[Re]: We edited the figure in accordance (see Fig. 1).

Figure 4. Could the switch from an alpha diversity mechanism to a beta diversity mechanism with nutrient addition be an artifact caused by the differences in scale? I would imagine that the 250m scale for the second experiment would have greater beta diversity which could influence the relative importance of the alpha vs. beta diversity in the SEM. In addition, figure S6 appears to show that adding nutrients increased beta diversity and lowered alpha diversity which could also explain the shift in relationships between the two SEM models.

[Re]: Thank you for pointing out this possible misunderstanding. First, we have modified the text to clarify (see lines 156-157) that both the pre-treatment (i.e. ambient

heterogeneity) and the post-treatment (i.e. increased heterogeneity) have exactly the same spatial scale (i.e. a set of local plots (25 m²) arranged in blocks that collectively represent a larger scale (an arrangement of 10 of those local plots resulting in 250 m²)). Second, as the reviewer noted, alpha diversity decreased and beta diversity increased under increased environmental heterogeneity (the decrease in alpha diversity may be a consequence of increased mean nutrient inputs instead of a consequence of increased heterogeneity). Despite these changes in mean values of both alpha and beta diversity, range values and variances did not change (see Fig. 4). To be sure, nevertheless, we centered and scaled alpha and beta diversity, and reevaluated the significance of the diversity*heterogeneity interaction using the scaled predictors. Below (see figure) we show the relationship between scaled diversity (alpha and beta) and spatial variability of biomass for ambient heterogeneity scenarios (in red) and increased heterogeneity scenarios (in skyblue). Inserted statistics are the results of the evaluation of the diversity*heterogeneity interaction significance. As these results are consistent with presented results, we are confident that they are not a consequence of changes in the predictor's distribution characteristics.

Reviewer #2 (Remarks to the Author):

In this manuscript, the authors tested the relationships between different levels of biodiversity (alpha, beta, and gamma diversity) and spatial variability in productivity and examined how increased environmental heterogeneity can alter these relationships. To do this, the authors used the dataset of a global network of experimental grasslands in which nutrients and herbivore exclusion were manipulated. They found that alpha and gamma diversity can reduce overall spatial productivity variability. They also found that beta diversity could either increase (when environmental heterogeneity is low) or decrease (when environmental heterogeneity is high) spatial variability in productivity. The study was well designed, and the statistical analysis used by the authors (LMMs and SEMs) and the interpretation of the results seems good to me. I have only a few minor comments that I hope can clarify the manuscript.

[Re]: Thank you, we appreciate these positive comments about our paper.

- In the title, "cancel" is a very strong word; I suggest replacing it with a weaker one. What "diversity" (alpha, beta, or gamma) are you referring to in the title, since increased heterogeneity has different effects on different scales of diversity?

[Re]: Thank you for pointing this out. To solve this problem we now entitled the paper "Environmental heterogeneity modulates the effect of biodiversity on the spatial variability of grassland biomass".

- I feel the introduction missed to tell readers why we should care about spatial variability. Does it affect ecosystem functioning or services when human activities increase or decrease spatial variability (productivity)?

[Re]: Thank you for this observation. We have now included text in the Introduction explaining the importance of spatial variability for the spatial reliability of ecosystem function and services (see lines 133-138): "Biodiversity loss at different scales⁵ is an important consequence of anthropogenic activities that also impacts the functioning of ecosystems. While biodiversity-functioning research has predominantly focused on temporal stability of biomass, less is known about spatial stability¹³. However, if biodiversity can buffer environmental change and stabilize spatial ecosystem functions and services, then biodiversity restoration and conservation will concurrently maximize functioning and spatial reliability³ in changing conditions."

- Both environmental heterogeneity and beta-diversity are spatially scale dependent. The experimental systems used in this study are still at the plot level (or a group of plots). The spatial change in both environmental heterogeneity and beta diversity across larger natural spatial scales might be very different from the system studied here. The author should also discuss the generality of the results at larger spatial scales. Discussing the scale of the space across which the study was performed is important as the authors also linked their results with biotic homogenization, which is also a spatial scale-dependent process.

[Re]: We thank the reviewer for pointing this out. We agree that it is important to discuss the implications of the scale-dependency of our results. In this new version we included the following text in the Discussion section (see lines 315-330): "The most likely driver of spatial heterogeneity at the spatial scale of our study design (i.e. hundreds of meters) is plot-scale variability of biotic or abiotic conditions. Spatial heterogeneity in environmental conditions is usually the result of concurrent, superimposed gradients occurring at multiple

spatial scales, or multiple disturbances interacting with each other³⁸. Biomass production often varies in response to this combination of coarse and fine-scale heterogeneity. Results of studies evaluating the effect of biodiversity on ecosystem function are often scale-dependent. For example, small-scale studies are more likely to be at the spatial scales at which niche-partitioning and competitive exclusion operate. Large-scale studies, on the other hand, are likely to detect the effects of site-scale factors (e.g., climate, herbivory) that may covary with diversity, thereby reducing the ability to detect niche partitioning and competition³⁹. At larger spatial scales, the importance of alpha diversity may decrease (niche partitioning becomes less important relative to extrinsic factors). Concurrently, the importance of beta diversity may increase (as different species are filtered into environmental conditions where their traits most efficiently convert resources into biomass)⁴⁰. Thus, even among the largest patches, diversity may continue to have an additional buffering effect on spatial variability in biomass production⁴¹.”

- In the study, increased environmental heterogeneity is treated as a binary variable (i.e. ambient heterogeneity and increased heterogeneity). The grouping is very coarse. The authors should also try to quantify the effect of environmental heterogeneity, maybe by grouping the blocks to different levels of environmental heterogeneity. I can of course understand that the number of the levels will be limited by sample size.

[Re]: We agree with the reviewer that having two levels of heterogeneity (ambient heterogeneity and experimentally increased heterogeneity) is coarse. Following reviewer suggestions we have added an additional analysis (as Supplementary Material) using 3 levels of environmental heterogeneity. This new analysis shows that the scenarios of intermediate heterogeneity (i.e., “moderately increased heterogeneity”) present intermediate slope values (see lines 218-219 and Supplementary Fig. 6).

As our experimental design has only two imposed levels of environmental heterogeneity, we statistically shuffled plots across blocks to create scenarios consisting in artificial “blocks” with “moderately increased heterogeneity”. For each of the 42 experimental sites, we grouped together 10 plots originally belonging to the 3 different blocks and collectively representing 4 different environmental conditions (e.g. the 3 control plots, the 3 +K μ plots, the 3 +P plots and 1 +PK μ plot). Given that this rearrangement allows only one artificial block per site (composed of plots from different blocks), we had to also create one block for the remaining two levels of heterogeneity by shuffling plots of different blocks. To create the (low) “natural heterogeneity” scenarios, we randomly sampled and grouped together 10 pre-treatment plots (from the 30 plots per site that resulted by pooling together the plots of the 3 blocks). Similarly, the “highly increased heterogeneity” scenario was created by randomly choosing one of the three replicates of each of the 10 treatments, leading to an artificial block with 10 plots (one per treatment) randomly chosen from any of the 3 original blocks. For each site and level of heterogeneity, we estimated alpha, beta and gamma diversity, as well as the spatial variability of biomass as described in the main text. We then modeled the relationships of the different scales of diversity with the spatial variability of biomass with type II regression and constructed confidence intervals for the slopes of the relations.

Thank you, again, for this suggestion. By grouping the blocks to different levels of environmental heterogeneity, your suggestion helps us to reinforce the role of the degree of heterogeneity controlling the slope of the relationship.

- L334: why 2750 patches from 271 larger scales?

[Re]: We now more clearly explained that not all sites present larger scales (blocks) composed of 10 patches (plots), as there are some exceptions that have larger scales with only 8 patches (see line 367 and Supplementary Table S1). We thank the reviewer for pointing this out, as by addressing the concern we detected that total patches are 2700 instead of 2750 (we have now corrected the number, see line 368).

- L341-343: I suggest here more explicitly describe the 10 patches (treatments) with +N, +P, +Fence etc as described in Borer et al. 2014.

[Re]: We made the change as suggested (see lines 375-380)

- The grasslands likely differ in the length of years across which they are sampled. Although this is more likely to have a consequence on temporal variability of the plots. But have the authors considered that this also has some consequences on the spatial variability. With increased sampling period, the plots in the same grassland are more likely to be affected by external climatic drivers (extreme climates for example) and homogenize the plots.

[Re]: We thank the reviewer for pointing this out. We now clarify that sites in our study have the same length of treatment years (see lines 394-396). As noted by the reviewer, although NutNet experimental design and data collection follow the same protocols in all participating sites (allowing global analyses on herbaceous-plant dominated ecosystems), sites differ in the length of years across which they are sampled. This is because initial NutNet sites started the nutrient addition treatment in 2008, but most sites started the experiment in the following years as new sites were incorporated into the network. We agree with the reviewer that this could affect our estimates of the spatial variability of biomass. However, in our analysis, we used pre-treatment (year 0) sampling data as well as post-treatment (4th year of treatment implementation) sampling data, so duration was the same at all sites.

- L369-370 is repeating L365. Consider simplifying it.

[Re]: We reordered and edited these sentences to avoid confusion. One sentence is the description of biomass sampling and the other sentence is the description of species cover estimation (see lines 403-415).

- L390: I do not agree that the three diversity indexes are independent. Gamma diversity is somehow a product of alpha and beta diversity.

[Re]: We agree and have edited the text in accordance (see line 431; and also the supplementary Table 5).

- L425: I am not familiar with this regression. But what do you mean by averaging values? Can you more explicitly specify this?

[Re]: We now more explicitly specify that this regression model does not allow inclusion of random structures in order to reflect our multi-level design (i.e. 83 sites, 3 blocks per site) and thus, to avoid pseudoreplication, we averaged the values of the 3 blocks per site for this analysis (see lines 467-472).

- L437-439: Instead of doing this way, why not just include both climatic variables and biodiversity as explanatory variables in the same model? And the authors can also check whether there is any interactive effect between climatic variables and diversity indexes.

[Re]: In this corrected version, we added the analysis suggested. Specifically, we performed a multi-model inference to select the simplest models that explained the most

variation (of spatial variability) based on Akaike's information criterion (AIC). Candidate models represented every possible combination of explanatory variables (i.e. the subset of bioclimatic variables along with the different scales of diversity) as well as the interactions between the bioclimatic variables and the different scales of diversity (see lines 484-489). We presented these results in a new Supplementary table (Supplementary Table 4).

- L448: "when appropriate" sounds very unclear. Please specify it.

[Re]: As we included sites as random effects for all individual models, we removed "when appropriate" from the sentence (see lines 497-498). We thank the reviewer for pointing this out.

- The authors should also deposit and share the codes and core dataset in one of public repositories, so the interested referee/readers can reproduce the results.

[Re]: All data and code for these analyses will be published and publicly available via EDI after this paper is accepted. Currently, the raw data and the complete R code that support our findings are available via GitHub (https://github.com/juanalberti/spatial_variability). In this revised version of the manuscript we added a Data availability statement and a Code availability statement (see lines 516-521).

- Fig. S1: why the authors hypothesize that there is no effect of environmental heterogeneity on gamma diversity.

[Re]: We indeed hypothesize that environmental heterogeneity affects (increases) gamma diversity. As, in our study, environmental heterogeneity is a measure of among-patches heterogeneity, we hypothesize that the effect of environmental heterogeneity on gamma diversity is not direct (see Supplementary Figure 7), but mediated by beta diversity (i.e. differences in environmental conditions among patches drive to species turnover; see Tamme, R., Hiiesalu, I., Laanisto, L., Szava-Kovats, R. & Pärtel, M. (2010). Environmental heterogeneity, species diversity and co-existence at different spatial scales. *J. Veg. Sci.*, 21, 796–801).

- L455: Do you mean Table S1 here?

[Re]: Thank you for detecting the error. We made the correction (see line 505).

REVIEWERS' COMMENTS

Reviewer #1 (Remarks to the Author):

I am satisfied with the Author's responses to my queries on the first round of review. The modification made in response are reasonable and I do not have further concerns.

Reviewer #2 (Remarks to the Author):

I made many comments on the previous version of this manuscript. The authors have made extra in-depth analysis according to my comments. I am pleased with these new changes and analysis and I thank the authors for their efforts to improve the manuscript. I have no further comments on this manuscript and therefore suggest acceptance of the paper. Congratulations!

Point-by-point responses to reviewers comments (authors replies in blue)

Reviewer #1 (Remarks to the Author):

I am satisfied with the Author's responses to my queries on the first round of review. The modification made in response are reasonable and I do not have further concerns.

[Re]: We appreciate your positive comments and helpful suggestions that greatly helped to improve the quality of our manuscript. We are truly grateful.

Reviewer #2 (Remarks to the Author):

I made many comments on the previous version of this manuscript. The authors have made extra in-depth analysis according to my comments. I am pleased with these new changes and analysis and I thank the authors for their efforts to improve the manuscript. I have no further comments on this manuscript and therefore suggest acceptance of the paper. Congratulations!

[RE] Thank you very much for reviewing our manuscript and for providing constructive comments. Your comments and suggestions have been very helpful and have greatly improved the strength and relevance of our analyses and conclusions.

Yours sincerely,

Pedro Daleo
on behalf of all coauthors